# Weak Spatial Target Extraction Based on Small-Field Optical System

**DOI:** 10.3390/s23146315

**Published:** 2023-07-11

**Authors:** Xuguang Zhang, Yunmeng Liu, Huixian Duan, E Zhang

**Affiliations:** 1Shanghai Institute of Technical Physics, Chinese Academy of Sciences, Shanghai 200083, China; 17860730760@163.com (X.Z.); lym_sitp@163.com (Y.L.); hxduan005@163.com (H.D.); 2Key Laboratory of Infrared System Detection and Imaging Technology, Chinese Academy of Sciences, Shanghai 200083, China; 3Hangzhou Institute for Advanced Study, Hangzhou 310024, China; 4University of Chinese Academy of Sciences, Beijing 100049, China

**Keywords:** small-field telescope, space target detection, image preprocessing, target signal enhancement, multi-frame projection, adaptive filtering

## Abstract

Compared to wide-field telescopes, small-field detection systems have higher spatial resolution, resulting in stronger detection capabilities and higher positioning accuracy. When detecting by small fields in synchronous orbit, both space debris and fixed stars are imaged as point targets, making it difficult to distinguish them. In addition, with the improvement in detection capabilities, the number of stars in the background rapidly increases, which puts higher requirements on recognition algorithms. Therefore, star detection is indispensable for identifying and locating space debris in complex backgrounds. To address these difficulties, this paper proposes a real-time star extraction method based on adaptive filtering and multi-frame projection. We use bad point repair and background suppression algorithms to preprocess star images. Afterwards, we analyze and enhance the target signal-to-noise ratio (SNR). Then, we use multi-frame projection to fuse information. Subsequently, adaptive filtering, adaptive morphology, and adaptive median filtering algorithms are proposed to detect trajectories. Finally, the projection is released to locate the target. Our recognition algorithm has been verified by real star images, and the images were captured using small-field telescopes. The experimental results demonstrate the effectiveness of the algorithm proposed in this paper. We successfully extracted hip-27066 star, which has a magnitude of about 12 and an SNR of about 1.5. Compared with existing methods, our algorithm has advantages in both recognition rate and false-alarm rate, and can be used as a real-time target recognition algorithm for space-based synchronous orbit detection payloads.

## 1. Introduction

Space targets include celestial bodies, in-orbit satellites, and space debris. As humans continue to launch aerospace equipment into space, more and more space debris is generated by space activities. Due to the long natural decay cycle, various orbital spaces outside Earth are becoming increasingly crowded. The US space surveillance network has catalogued over 40,000 space targets, but there are still many smaller pieces of space debris that are difficult to observe [1]. This space debris is usually small and very dim. If a normally operating spacecraft collides with space debris, space traffic accidents will occur and cause damage to or derailment of space equipment [2]. Identifying and locating weak space debris is of great significance for ensuring the safety of the space environment and spacecraft.

Space target monitoring methods mainly include radar technology and optoelectronic technology. Radar technology belongs to active detection, which has the advantage of uninterrupted operation throughout the day. But the limited detection distance and accuracy make it difficult to identify small space debris in high orbits by radar. The advantages of optoelectronic technology are the long detection distance and high detection accuracy, which can be used for synchronous orbit debris detection [3]. But it is easily affected by weather due to its passive detection mode. Space target detection modes can be divided into ground-based and space-based detection. Usually, different detection methods are combined to form a space surveillance network. The relatively mature technology is the advantage of ground-based detection systems, but they are susceptible to geographical limitations and atmospheric interference [4]. Space-based detection systems operate outside the atmosphere, so they can directly receive target radiation energy without passing through clouds, and can detect a wider spatial range [5]. Space-based detection systems have significant advantages over ground-based systems, and therefore will be the main development direction in the future. The algorithm proposed in this article can be applied to the optoelectronic detection of space-based synchronous orbits.

Due to the inherent detector noise, cosmic radiation noise, and stray light noise generated by interference sources, weak targets usually have a very low signal-to-noise ratio(SNR) in images. In addition, due to weak targets often being small and far away from the camera, they only occupy few pixels and do not have obvious geometric texture features. In cosmic space, with the increase in stellar magnitude, the number of stars increases exponentially. Therefore, weak spatial targets have extremely complex backgrounds. The real-time extraction of spatial targets in complex backgrounds with low SNR is currently a research difficulty.

The best method for detecting small targets in sequence images is the three-dimensional matching filter, which can maximize the target SNR when the motion information is known [6]. However, when prior information on the target is unknown, it requires traversing all paths in all images, which requires a huge computational load and cannot achieve real-time processing. To balance the real-time capability and reliability of detection, many algorithms have emerged in recent years attempting to identify small targets as accurately as possible while reducing computational complexity.

Reference [7] proposes a star suppression method based on multi-frame maximum projection and median projection, but there will be residual stars after processing, which are difficult to distinguish in the case of weak targets. Reference [8] proposes a star detection method of multiplying adjacent frames after registration, but it is difficult to detect weak stars when registration is inaccurate. Reference [9] proposes a star suppression method based on enhanced dilation difference, which solves the problem of edge residue caused by inaccurate registration and star brightness changes, but it may lose weak targets near the star. The above methods are essentially single-frame target extraction after fusing several frames of images. Due to the inevitable information loss during the fusion process, the reliability of target recognition is limited.

Reference [10] proposes a spatial object detection algorithm based on robust features, but it is not suitable for weak targets with only a few pixels. References [11,12] use guided filtering to remove stars and identify target motion fringes, but it is difficult to directly generate stable and continuous target fringes in high-orbit small-field spatial object detection. References [13,14] use deep learning algorithms to classify targets, but currently the model can only distinguish target motion stripes and point noise under different SNRs. Reference [15] proposes a spatial object detection method based on parallax, but this method requires two imaging devices and can only distinguish between targets and stars in synchronous orbits. The multi-level hypothesis testing method proposed in references [16,17] predicts the conversion region through inter-frame motion, which can be used to detect continuous or discontinuous target trajectories. However, it is inaccurate and slow in multi-target detection. References [12,18] use Hough transform to detect target trajectories, but it requires detecting and removing stars in advance. References [19,20] use two-dimensional matched filtering to detect targets, but this method requires traversing paths in different directions, lengths, and shapes in images. Although the computational complexity is smaller than that of three-dimensional matched filtering, it still takes a long time.

Most existing algorithms recognize space targets with wide-view telescopes, but to improve the detection ability for weak targets, small-field telescopes need to be used to improve spatial resolution. This article proposes a method of weak-target extraction from small-field starry backgrounds based on spatiotemporal domain adaptive directional filtering, which achieves low SNR multi-spatial target recognition in small-field ground-based optical systems. Due to the complex starry background, identifying stars is a necessary operation before recognizing space debris when both are imaged as dots. Therefore, this article analyzes the extraction of weak stars from images, providing necessary technical guarantees for space debris recognition. The overall idea of the article is as follows: Firstly, the bad points are repaired and the non-uniform background in the image is suppressed. Secondly, enhancement algorithms are used to improve the SNR of weak stars. Thirdly, key information is compressed and extracted from the sequence images with a multi-frame projection algorithm, and an iterative thresholding algorithm is used for thresholding processing. Then, the registration results are used to construct directional filter operators, which retain trajectory features while filtering out noise. Afterwards, different morphological operators are used to perform morphological closure operations to connect the target trajectories. Next, adaptive median filter operators are used to eliminate false trajectories and obtain the final results of star trajectories. Finally, the trajectories obtained from multi-frame projections are deprojected to obtain star detection results for a single image.

The method proposed in this article can effectively detect weak stars in images. The maximum detectable magnitude is greater than 11 when using ground-based telescopes. If space-based synchronous orbit detection is used, the relative speed between the stars and camera is same as that of ground-based detection. In addition, space-based detection is significantly less affected by noise interference compared to ground-based detection, and weaker spatial targets can be detected under the same conditions. Therefore, the method proposed in this article has important reference significance for the real-time detection of weak targets in space-based synchronous orbits, spacecraft navigation, and space debris identification and cataloging. The structure of this article is as follows: The first section introduces the research background and current situation. The second section introduces weak star recognition methods. The third section introduces the experimental results. The fourth section summarizes the entire text.

## 2. Target Recognition

The small-field large aperture telescope can improve the angular resolution of the imaging system and fully collect photons. To improve the target imaging SNR while avoiding electron overflow caused by bright stars, we set the camera integration time to 1 s. The motion speeds of high-orbit space debris and background stars in the image are both slow, so they are imaged as point targets.

After analyzing the global histogram (Figure 1a) and local histogram (Figure 1b) of the taken image, it is found that most pixels are distributed between 14,000 and 17,000 and roughly obey the Gaussian distribution. According to the local grayscale histogram, it can be seen that the pixel number on right side of the peak is more than that on left side, because there are weak spatial targets submerged in Gaussian noise. If threshold segmentation is directly used, it is easy to eliminate weak targets and mistakenly identify some noise as targets. Identifying these weak targets that are difficult to distinguish from noise is the significance of this article.

### 2.1. Bad Point Repair

The detector-inherent bad points are usually caused by defects in the photoelectric conversion array. Morphological operators are structural elements with a specific size and shape which change the grayscale by locally matching structural elements with the image. In order to clearly observe bad points, this article uses large-size morphological operators (10 × 10) for corrosion operations. Then, it can be seen that there are several black points in the image. Black spots are local pixel anomalies caused by detector defects, which are amplified after morphological corrosion operations. Figure 2a,b, respectively, show the original image and the morphological corrosion image. It can be seen that the grayscales of different bad points are not same. Some points are completely damaged, causing the pixel grayscale to be zero, while others have small grayscale due to low photoelectric conversion efficiency.

The idea of the bad point repair algorithm used in this article is as follows: If the grayscales at the same position in multiple consecutive frames are significantly abnormal compared to the neighborhood, then the position is determined to be a detector bad point, and the grayscale is set to the mean of its neighborhood. Figure 2c,d, respectively, show local enlarged images before and after bad point repair.

### 2.2. Background Correction

Due to the weak stray light in the cosmic environment and the non-uniform response of the detector, the starry image background is not uniform. Although it is difficult for humans to directly observe non-uniformity with their eyes, its impact on weak-target recognition is significant. To observe the non-uniform background more intuitively, Figure 3a displays the image in a partial grayscale range. It can be seen that the image edges are severely affected by stray light, and there are also striped backgrounds formed by non-uniform response.

Morphological operation is a commonly used method for detecting backgrounds [21]. But when the operator size is small, some larger stars will be mistaken for the background. If the size of the operator is increased, although the above phenomenon will decrease, the background estimation will be inaccurate, which is unacceptable in small-target detection. To improve the performance of small-size operators, this paper proposes an improved morphological background estimation algorithm. After obtaining the preliminary background through morphological operation, the threshold is determined based on its mean and variance. Then, the grayscale which is greater than the threshold is replaced with the mean of its neighborhood. After correcting the background through the above operation, it is subtracted to eliminate non-uniformity. This method takes the characteristics of slow changes in non-uniform noise and corrects background areas with excessively large grayscales. Figure 3b shows the image after non-uniformity correction, © shows the three-dimensional image of preliminary estimated background, and (d) shows the three-dimensional image of the final estimated background.

### 2.3. Image Enhancement

In low-orbit target detection, due to the fast motion speed of targets, a camera can usually obtain target motion fringes within the integration time. Currently, many works use the motion information contained in fringes to identify the target. However, in high-orbit target detection, the target’s movement is slower and usually point targets can only be obtained within a limited integration time.

When detecting weak targets, there is a significant difference in energy between bright and dark stars. When the exposure time is long, CCD will easily cause electron overflow around bright stars, leading to overexposure. This will create bright stripes on the image and cause significant interference in target recognition. Figure 4 shows the overexposed image after extending the integration time. Therefore, it is difficult for cameras to increase the integration time too much to detect weak targets.

The image used in this article has an integration time of one second. The probability of overexposure within a one-second integration time is low. The synchronous orbit radius r = 42,157 km, gravitational constant $G=6.67\times 10^{-11}N\cdot m^{2}/{kg}^{2}$, and earth mass $M=5.965\times 10^{24}$ kg are included in the universal gravitation formula to calculate the synchronous orbit satellite angular velocity $w=\sqrt{r^{3}/G\cdot M}=4.17\times 10^{-3}{}^{\circ}/s$. So, during the integration time, stars will not fly out of the camera’s field. But in star tracking mode, the telescope needs to be moved.

SNR is defined as the ratio of mean signal to noise standard deviation. Within a certain time range, the longer the camera exposure time, the higher the weak-target SNR in the image. But when there are target dragging or streaks in the image, the target SNR will no longer increase and gradually decrease, because at that time, the noise energy accumulates but the signal energy remains unchanged.

To improve the SNR of weak targets, this article uses a multi-frame image enhancement algorithm. Utilizing the principle of correlated target signals but uncorrelated noise signals after registration, multiple frames of images are accumulated. After accumulating N frames, the target signal is increased N times, and the noise variance is increased N times. According to the SNR calculation in Formula (1), the total SNR is increased $\sqrt{N}$ times [22]. Figure 5 shows the ratio curve of noise power between the N-frame-enhanced image and the original image.
(1)$${SNR}_{N}=\left(N\cdot {SNR}_{0}\cdot \sigma_{0}\right)/\sqrt{N\cdot {\sigma_{0}}^{2}}=\sqrt{N}\cdot {SNR}_{0}$$

Because the signal readout time is much shorter than the exposure time, we assume that the frame interval is equal to the exposure time. If the target motion speed is v (pixel/s), the target SNR reaches maximum when the camera integration time is 1/v. It is assumed that the optimal signal/noise ratio is SNR, and the noise variance is $\sigma_{0}$. If the camera integration time is t, which is larger than 1/v, the signal/noise ratio of the stripe target is calculated with Equation (2). If t is smaller than 1/v, the signal/noise ratio of the point target is calculated with Equation (3).
(2)$${SNR}_{0}=\left(SNR\cdot \sigma_{0}\right)/\sqrt{t\cdot v\cdot {\sigma_{0}}^{2}}=SNR/\sqrt{t\cdot v}$$
(3)$${SNR}_{1}=\left(t\cdot v\cdot SNR\cdot \sigma_{0}\right)/\sqrt{t\cdot v\cdot {\sigma_{0}}^{2}}=\sqrt{t\cdot v}\cdot SNR$$

When using (1/t·v) frames to enhance the point target, the SNR increased by $1/\sqrt{t\cdot v}$ times compared to the original image, which can achieve the optimal SNR. When the number of frames used for enhancement is more than (1/t·v), the enhanced target SNR will be better than the optimal target SNR of a single frame. From this, it can be seen that the target SNR after the enhancement of a multi-frame short exposure image may be better than the target stripe SNR obtained by a long exposure time.

Considering that the more images that are used for enhancement, the smaller their common sub-images will be, and also balancing the real-time capability and reliability, this article selects five consecutive frames for enhancement. The five registered images are combined to obtain the target enhancement image. Due to the small variation, the intuitive changes in the image are difficult to observe. From the grayscale histogram shown in Figure 6, it can be seen that the noise variance in the enhanced image is significantly reduced compared to before, and more small targets are no longer submerged in the noise.

### 2.4. Multi-Frame Projection

Through the analysis in Section 2.4, it can be concluded that short-exposure point targets may have a better SNR than long-exposure stripe targets after multiple-frame enhancement. However, point targets do not have the motion information contained in striped targets, which makes it difficult to accurately extract them from noisy images even with a high SNR. In order to quickly extract target motion information, this article uses multi-frame projection to fuse information from several images so that the target forms stripes in the image, which is conducive to extracting weak targets from noise.

Reference [23] proposes the optimal projection operator, which is derived from the maximum likelihood ratio. The operator expression (4) is as follows. Where $z_{0}$ is the pixel value after projection, N is the number of projected frames, $r\left(i,j,t\right)$ is the grayscale value of a certain frame at position $\left(i,j\right)$, and S is the SNR of this point. Due to the need to estimate the SNR of each point, which requires estimating the mean and variance of the background around the point, a significant amount of computation is required during projection.
(4)$$z_{0}\left(i,j\right)=ln\;[\sum_{t=1}^{N}e^{S\cdot r\left(i,j,t\right)}]$$

The full-dimensional matched filter sums the pixel grayscale in full-dimensional space along the specific velocity to detect moving objects in multiple frames, while the two-dimensional matched filter only detects targets with a specific size and shape in a single image. Reference [24] points out that when the length of the signal trajectory is M and the signal/noise ratio of the target signal is SNR, if the number of projected frames is N, then the output signal/noise ratio of the full-dimensional matched filter is $\sqrt{M}\cdot$ SNR, and the output signal/noise ratio of the two-dimensional matched filter after summation projection is $\sqrt{M}\cdot$ SNR/$\sqrt{N}$. It is difficult to obtain an analytical solution for the output signal/noise ratio after optimal projection and maximum projection. The output SNRs after projection are all lower than that of full-dimensional matched filtering, which is an inevitable result of partial information loss after multi-frame information fusion. When the SNR of the original image is low, the output SNRs of optimal projection and summation projection are similar, and the losses are both large. When the SNR of the original image is large, the output SNRs of optimal projection and maximum projection are similar, and the losses are both small. Equation (5) is the summation projection expression and Equation (6) is the maximum projection expression, where K is the number of projection frames.
(5)$$z_{1}\left(i,j\right)=mean\left[r\left(i,j,k\right)\right],(1\leq k\leq K)$$
(6)$$z_{2}\left(i,j\right)=max\left[r\left(i,j,k\right)\right],(1\leq k\leq K)$$

As we used five consecutive images in the target enhancement module, the output SNRs of maximum projection and average projection of five frames are quantitatively analyzed. Because of the high computational complexity, the optimal projection is discarded. Using five unrelated noise images that follow the normal distribution $\left(\mu ,\sigma \right)$, we statistically analyze the noise distribution after maximum projection. The experiment shows that it roughly follows a normal distribution (μ + 1.16σ,0.66σ). Figure 7 shows the grayscale histogram of the noisy image before and after projection.

Assuming the target grayscale is x, then Equation (7) is the expression for the target SNR before projection, and Equation (8) is that after maximum projection. According to the formula, when x > (μ+3.4σ), the target SNR after processing is improved compared to the original image. So, when using the maximum projection algorithm, if the target SNR is large, it can be further improved. If using the summation projection, then Equation (9) is the expression for the processed target SNR. According to the formula, the image SNR after processing is always smaller than that of the original image. After comparing the two projection methods, it can be concluded that when x > $\left(\mu +2.37\sigma \right)$, the target SNR after maximum projection is higher.
(7)$${SNR}_{2}=\left(x-u\right)/\sigma$$
(8)$${SNR}_{3}=\left(x-\left(\mu +1.16\sigma \right)\right)/\left(0.66\cdot \sigma \right)$$
(9)$${SNR}_{4}=\left(\left(x-u\right)/5\right)/\left(\sigma /\sqrt{5}\right)$$

The appropriate projection method can be selected based on the target SNR. The remaining part of this article selects the maximum projection method, which also has the advantage of conveniently releasing projection, because only the pixel frame index of the projected image needs to be recorded. Releasing the projection will play a significant role in target localization. Figure 8a,b, respectively, show the results of mean and maximum projection using five frames.

### 2.5. Threshold Processing

The threshold segmentation algorithm is used to eliminate unwanted background grayscale and residual noise. The commonly used threshold algorithms include the maximum inter class variance method, the iterative threshold method, and the adaptive threshold method [25]. But these methods only work well when there are two peaks in the histogram. In the grayscale histograms of starry images, there are usually single peaks composed of noise, and the target is scattered irregularly on larger grayscale values without forming peaks. This article improves on the traditional adaptive threshold method and proposes an iterative adaptive threshold algorithm.

First, the mean $\left(\mu \right)$ and variance $\left(\sigma \right)$ of the starry image are calculated, and $\left(\mu +3\sigma \right)$ is used as the threshold for preliminary threshold processing. Then, the mean $\left(u_{1}\right)$ and variance $\left(\sigma_{1}\right)$ of grayscale below the threshold are counted, and the threshold is processed again using $\left(u_{1}+3\sigma_{1}\right)$. After, the threshold is iteratively calculated in this way. According to the 3σ-Criterion, the iteration is stopped when the difference between new and previous threshold is less than 0.15%. The noise in the stellar image roughly obeys normal distribution, but there are also targets in the image. So, the background variance obtained from the first estimation is relatively large, and the value distributed within $\left(\mu -3\sigma ,\mu +3\sigma \right)$ is relatively high. After multiple iterations of estimation, the estimated background will gradually meet the 3σ guidelines.

The results from using the iterative threshold method to process starry images are shown in Figure 9. The threshold remains basically unchanged after approximately three iterations. After successfully estimating the background variance $\left(\sigma_{0}\right)$ in the original image, the image is processed with $\left(u_{0}+n\sigma_{0}\right)$ as the threshold, where $u_{0}$ is the mean of the projected image. The value of n determines the SNR of detectable targets in the original image. If n is too large, it is not conducive to weak-target recognition. If n is too small, it increases the false-alarm rate and computational burden.

### 2.6. Directional Filtering

To reduce the computational burden, the camera’s image acquisition rate is set to 2 s/frame, which makes some trajectories discontinuous. In addition, noise interference also makes it difficult to extract trajectories. In response to the above difficulties, this article proposes an adaptive directional filtering method. The specific steps are as follows. Firstly, an adaptive filtering operator is used for neighborhood submaximum filtering to preliminarily reduce noise. Then, the adaptive morphological operator is used to connect trajectories. Finally, adaptive median filtering is used to eliminate false trajectories and residual noise.

According to the method described in Section 2.5, we use 1.2 as the value of n to threshold the projected image. After threshold processing, continuous or discrete target motion trajectories are formed, while false targets and noise appear as points. Median filtering belongs to nonlinear filters and is a commonly used noise removal algorithm. However, when processing multi-frame projection images, the median filtering algorithm is prone to destroying trajectories. Reference [12] proposes an improved median filtering algorithm, while reference [13] proposes a local threshold filtering method. These methods have improved compared to traditional median filtering in processing starry image, but there are still some shortcomings.

In this paper, an adaptive filter operator is proposed. Based on the results of image registration and the distance from the points to the ideal straight line, the best operator can be found. The specific implementation method is as follows. Taking the filtering operator center as fixed point and calculating the slope based on the registration result, an ideal straight line is made within the filtering operator. Then, the distance from all pixels to the ideal line is calculated in the filtering operator, and m pixels with shorter distances are selected to form the filtering operator. Figure 10a shows the construction principle of the 5 × 5 filter operator, and the size and effective pixel number (m) of the operator can be set as needed.

In this article, C is defined as the maximum translation parameter of adjacent frames. Taking (6C + 1) as the adaptive filtering operator size, which can ensure that there are discontinuous breakpoints, the trajectory will not be damaged. The effective pixel number within the filtering operator is set to 2 (6C + 1). The filtering value is taken as the second largest value in the neighborhood, which increases the filtering stability. Figure 10b is the filtering operator used in this article. Logic and operation are performed on the images before and after adaptive filtering to obtain the filtering results. The above filtering process is repeated until the image grayscale no longer changes. If the original projection graph is M and the filtering operator is F, then Equation (10) is the calculation expression for the directional filtering graph (DF), where ∗ is the filtering operation.
(10)$$DF=M,DF=loop(\left(DF*F\right)\&DF)\;until\left(DF=loop(\left(DF*F\right)\&DF)\right)$$

Figure 11a,b show the local enlarged images before and after filtering. After using the method proposed in this paper, the trajectory features were successfully preserved while filtering out isolated noise. To demonstrate the superiority of our method, we compare it with improved median filtering algorithms and local threshold filtering methods. Figure 11c shows the image obtained after improved median filtering, with a small amount of target trajectories being damaged. Figure 11d shows the image obtained after local threshold filtering. Although it can retain most trajectory information while filtering out noise, it still has a significant impact on discontinuous trajectories. Our algorithm has significant advantages.

### 2.7. Trajectory Detection

After filtering out isolated noise points, the trajectory is connected using a large expansion operator and a small corrosion operator. All morphological operators are constructed using the adaptive method described in Section 2.6. The size of the expansion and corrosion operators are both (2C + 1), which can connect the broken line and ensure that the line length before and after processing is the same. The effective pixel number of the dilation operator is set to 3 (2C + 1), and the effective pixel number of corrosion operator is set to (3C + 1). A thinner corrosion operator can avoid line breakage. Equation (11) is the calculation expression for the trajectory connection image (CT), where A is the expansion operator and B is the corrosion operator, and ⊕ is the expansion operation and ⊝ is the corrosion operation. Figure 12a,b, respectively, show the morphological dilation operator and corrosion operator used in this paper.
(11)CT=DF⊕A⊝B

Figure 13a,b, respectively, show the enlarged images using adaptive morphological operators of expansion and post-expansion corrosion. All trajectories in Figure 13b are connected after morphological closure operations.

After connecting trajectories through morphological closed operations, iterative adaptive median filtering is used to remove false trajectories and residual noise, resulting in continuous trajectory detection results. The median filtering operator size is (4C + 1), which can filter out false trajectories that are too short. The effective pixel number within the operator is 2 (4C + 1). The median filtering used in this article does not take the median value, because a better effect is achieved when taking the value at 2/3 positions sorted from smallest to largest. Logic and operation are performed on the images before and after adaptive median filtering to obtain the filtering results. The above filtering process is repeated until the image no longer changes. Equation (12) is the calculation expression for the median filtering image (MF), where ∗ is the median filtering operation and C is the median filtering operator. Figure 14a shows the local adaptive median filtering graph. Equation (13) performs logic and operation between the median filtering image (MF) and the directional filtering image (DF) to obtain the final trajectory image (FT). Figure 14b shows the local final trajectory detection graph.
(12)$$MF=CT,MF=loop\left(\left(MF*C\right)\&MF\right)\;until(MF=loop\left(\left(MF*C\right)\&MF\right))$$
(13)FT=MF&DF

### 2.8. Target Positioning

This article applies a deprojection operation for the maximum projection image to extract stars from single frame. Given the maximum projection image $z_{2}\left(i,j\right)$, the release projection image ($R_{i}$) of each frame can be calculated. Equation (14) is the calculation method for the release projection image of the fifth frame. The grayscale of each pixel at the same position is compared between the maximum projection image and the fifth frame image. If they are the same, it indicates the projected image points are from the fifth frame image, and they are assigned to the release projection image of the fifth frame. Equation (15) is the calculation expression of the positioning image ($L_{5}$). By performing dot product operation on the trajectory image (FT) and the fifth frame release projection image ($R_{5}$), the star positioning image of the fifth frame can be obtained. Figure 15 shows the star positioning image and its partially enlarged image.
(14)$$R_{5}\left(i,j\right)=\left\{\begin{array}{cc} z_{2}\left(i,j\right) & ,r\left(i,j,5\right)=z_{2}\left(i,j\right)\\ 0 & ,r\left(i,j,5\right)\neq z_{2}\left(i,j\right) \end{array}\right.$$
(15)$$L_{5}\left(i,j\right)=R_{5}\left(i,j\right)\cdot FT\left(i,j\right)$$

## 3. Experimental Result

In this section, we analyze the recognition rate, false-alarm rate, and limit detection performance of the proposed method through experiments. The self-developed optical system used in the experiment has a field of 3.17° × 3.17°, 240 mm focal length, and 150 mm aperture. The detector uses e2v CCD47-20, which has 1024 × 1024 pixels, a 13.3 mm × 13.3 mm focal plane, 13 um pixel size, 11.18″ pixel resolution, and 16 bit AD quantization during image readout. Image processing is carried out in MATLAB R2017b, the computer processor is Intel (R) Core (TM) i5-7300HQ CPU(4-core, 2.5 GHz) with 8 GB memory. On the existing experimental platform, the algorithm can complete the operation within 4 s.

We used the Hipparcos catalog to test the recognition rate and ultimate magnitude recognition ability of our algorithm. Although the catalog contains stars that have magnitudes up to 12, it cannot be used to calculate false-alarm rates, because it only has complete stars with magnitudes of 9. We use the triangle-matching algorithm [26] to match starry images without prior information. If the matching fails, the points are reselected. When the matching results are not unique, the fourth point is selected and the pyramid-matching algorithm is used [27]. If the matching is successful, the attitude matrix is calculated based on the matching results, and the right ascension and declination of all the stars in the image are located. Then, all the stars are identified by matching with the Hipparcos catalog. The stars used for matching are marked in Figure 16a. The successfully identified stars are marked in Figure 16b with a recognition accuracy of 0.01°. Due to the flipped imaging of the camera, Figure 16 has been corrected compared to the original taken image.

Based on the camera pointing, camera FOV, and the star catalog, we created the ideal image, in Figure 17a, which only contains stars. The image rotation was calculated by using the image registration algorithm, and Figure 17b was obtained after correcting the ideal image. Because of the image rotation difference between the ideal and actual image, the two images differ slightly at the edge. Except for the M37 nebula in the upper part of the image, which is difficult to identify due to centroid positioning errors, other stars in the Hipparcos catalog are successfully identified. The success rate of single-frame recognition reaches 96.3%.

In Figure 18a, two stars with magnitudes higher than 11 are marked. They are numbered hip-27066 and hip-27825 in the Hipparcos catalog. They have approximately 16,000 grayscale and 1.5 SNR in the image. These two are the weakest stars in the Hipparcos catalog within this sky region, occupying less than 10 pixels, and both have been successfully identified. Figure 18b is the partially enlarged image of Figure 18a.

Due to the lack of a complete star catalog, it is difficult to use the star-matching method to verify the false-alarm rate of our algorithm. In this section, we associate the star extraction results from adjacent frames. After removing new stars that have entered and exited, if a certain star does not have a mapping relationship between adjacent frames, the detection result is regarded as a false alarm. The mapping criteria between two stars are designed as follows. Their centroid error is less than 1 pixel, the size error is less than 20%, and the brightness error is less than 20%. The statistical method for the false-alarm rate is not entirely reliable and is only for reference. Table 1 shows the detection results of stars with different SNRs. To demonstrate the superiority of the proposed method, we used the median projection method (MP) and the inter-frame multiplication method (IFM) to detect stars within the same region. The detection results of stars with different SNRs are recorded in Table 2 and Table 3. Indicators include recognition rate, false-alarm rate, and the average target number extracted from sequence stellar images.

Our method not only has a lower false-alarm rate and higher recognition rate compared to existing algorithms, but also has advantages in star brightness extraction and size extraction. The stars detected by MP have different brightness levels from the actual stars, which means that stars cannot effectively be removed from the original image. When the star brightness changes, the stars detected by IFM are prone to deformation, which will cause significant errors in centroid positioning.

## 4. Conclusions

In this article, we proposed a star detection algorithm based on small-field telescopes, which can achieve the real-time detection of stars with an SNR of 1.5 or higher, and its performance meets the requirements of space surveillance systems. With improvements in the detection ability of optical systems, the number of background stars in the image increases exponentially. When space debris and stars are both imaged as point targets, it is difficult to distinguish between them. Therefore, high-precision star detection is a prerequisite for space debris recognition.

The following conclusions can be summarized in this article. Our proposed background correction algorithm can maintain target energy while removing stray light. The multi-frame enhancement algorithm can increase the target SNR to no less than that obtained from long-exposure imaging. The multi-frame projection algorithm can compress multi-frame information and extract motion trajectories. The iterative adaptive threshold algorithm can accurately estimate background parameters. The adaptive filtering algorithm we proposed can remove noise while preserving the target trajectory, and has high robustness to trajectory breakage and overlap. After the adaptive morphological algorithm connects trajectories, the adaptive median filtering algorithm can filter out false trajectories. Finally, releasing the projection operation can accurately locate stars in a single image.

The actual starry image processing result showed that the proposed method can overcome the difficulties of star extraction in complex backgrounds with small fields, and has a high detection rate and low false-alarm rate. Compared to wide-field long-exposure imaging, it has the advantages of high real-time performance and strong detection ability. The method proposed in this article has important reference significance for the real-time detection of weak stars in space-based synchronous orbits, spacecraft navigation, high-precision cataloging of space debris, and so on.

## Figures and Tables

**Figure 1 sensors-23-06315-f001:**
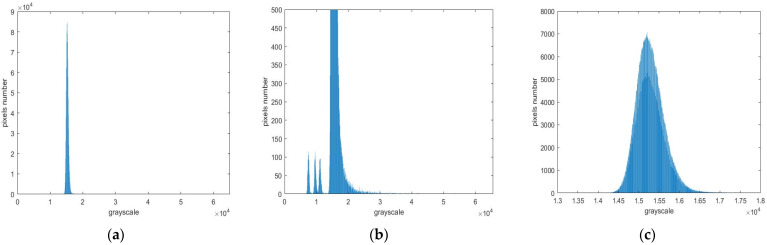
Image grayscale histogram. Panel (**a**) is the original image. Panel (**b**) is the enlarged global histogram of the original image. Panel (**c**) is the local histogram of the original image.

**Figure 2 sensors-23-06315-f002:**
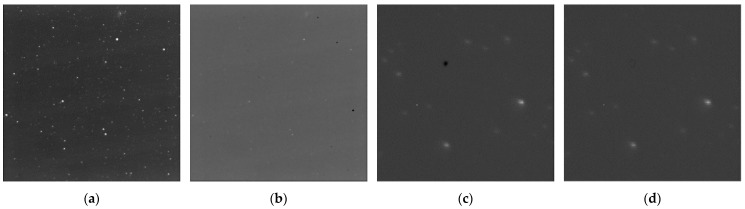
Panel (**a**) is the original image. Panel (**b**) is the result of morphological erosion by a large operator on the original image. Panel (**c**) is a local enlarged image of a bad point in the original image. Panel (**d**) is the processed local enlarged image of bad point.

**Figure 3 sensors-23-06315-f003:**
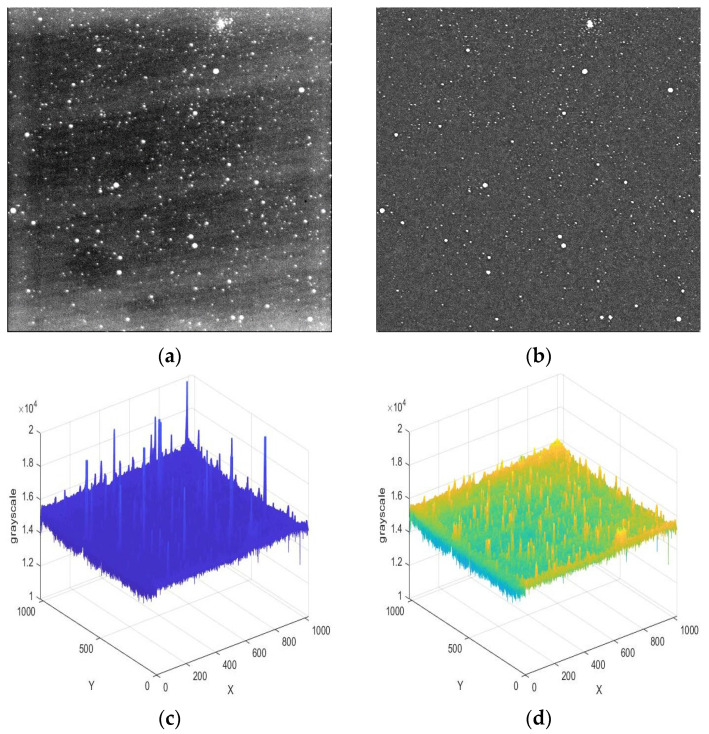
Panel (**a**) is the original image after adjusting the grayscale display range. Panel (**b**) represents the image obtained from background correction. Panel (**c**) represents the first estimated background. Panel (**d**) is the final estimated background.

**Figure 4 sensors-23-06315-f004:**
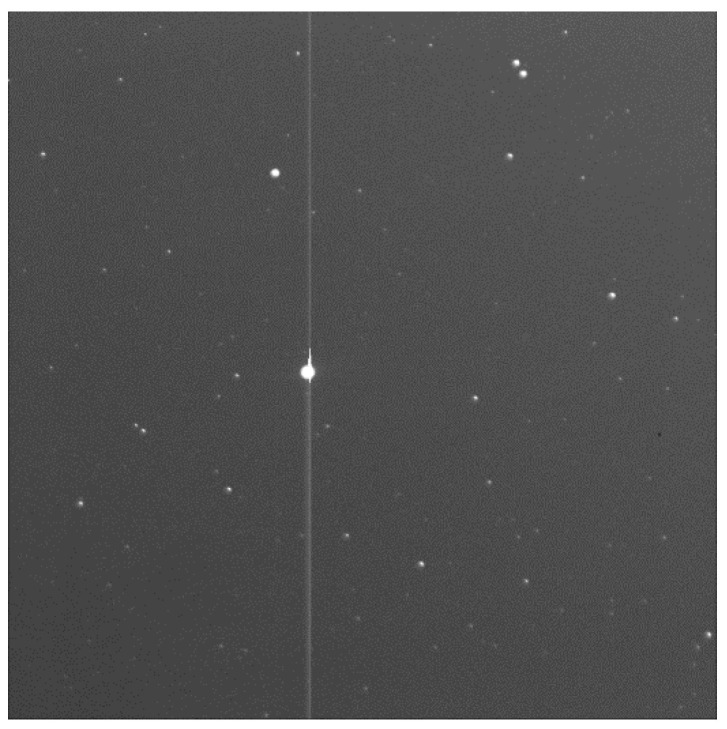
Excessive stellar energy leads to electron overflow after extending the integration time.

**Figure 5 sensors-23-06315-f005:**
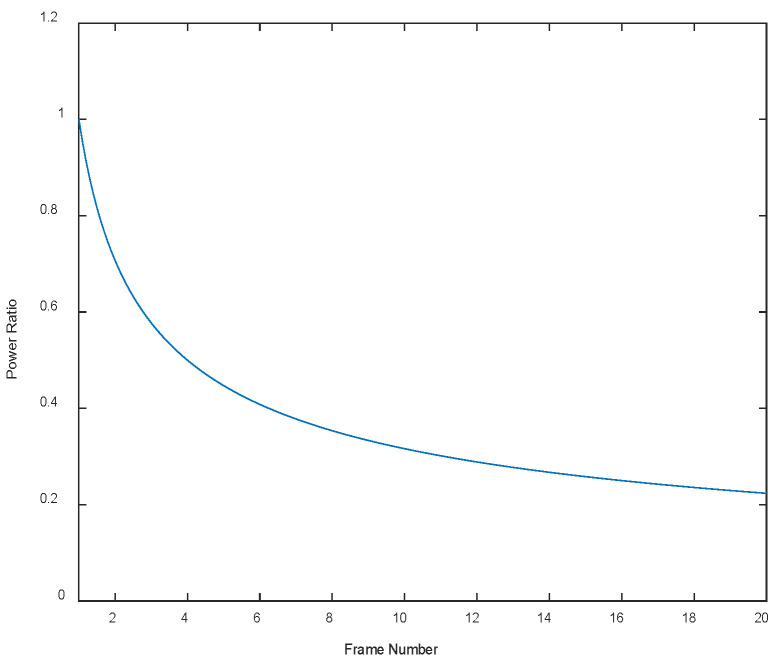
The ratio of noise power between the original image and the enhanced image using N frames.

**Figure 6 sensors-23-06315-f006:**
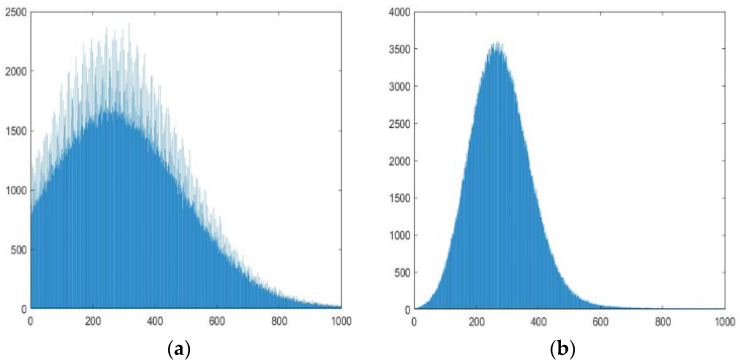
Panel (**a**) represents the local grayscale histogram of the preprocessed image. Panel (**b**) represents the local grayscale histogram of the enhanced image.

**Figure 7 sensors-23-06315-f007:**
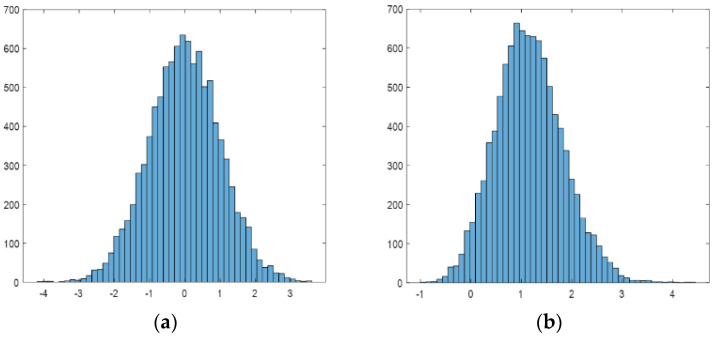
Panel (**a**) is the statistical histogram of standard normal distribution, and panel (**b**) is the statistical histogram after the projection of the maximum value of five frames.

**Figure 8 sensors-23-06315-f008:**
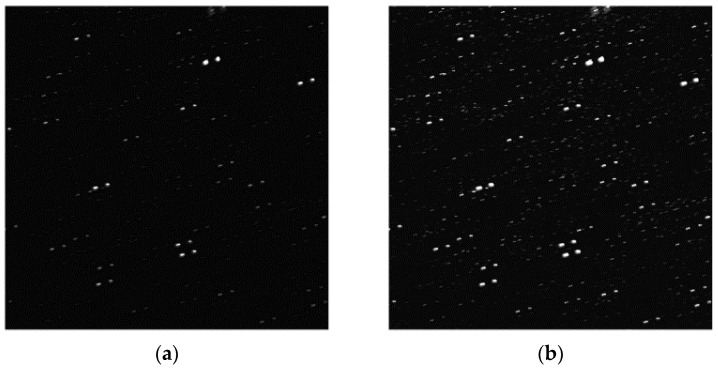
Panel (**a**) represents the mean projected image using five frames. Panel (**b**) represents the maximum projected image using five frames.

**Figure 9 sensors-23-06315-f009:**
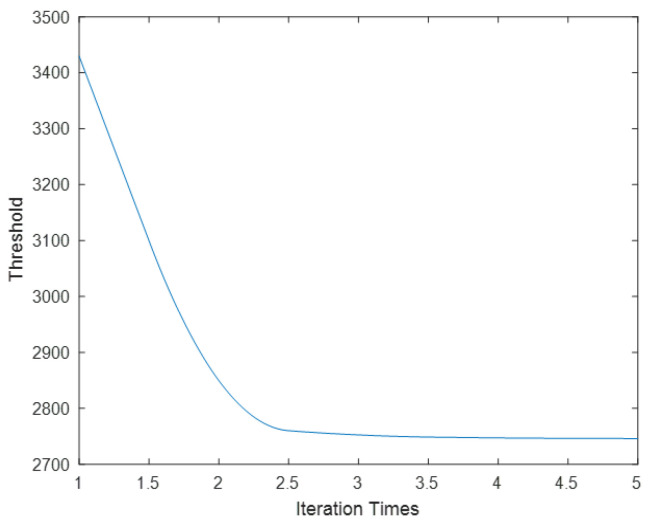
The variation in segmentation threshold with number of iterations.

**Figure 10 sensors-23-06315-f010:**
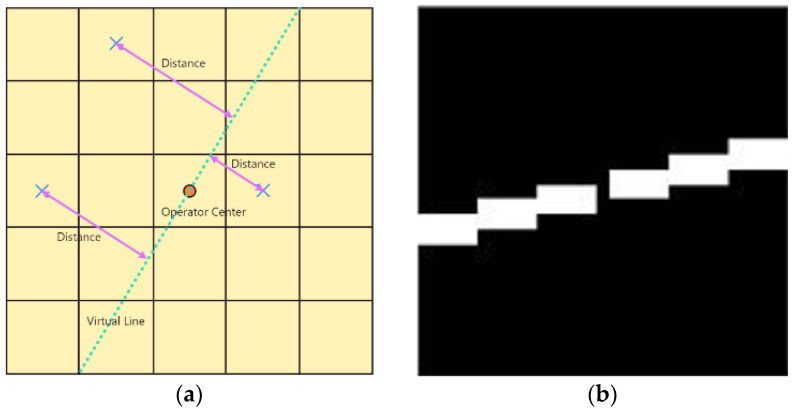
Panel (**a**) is schematic diagram of constructing an adaptive filter operator with size of 5 × 5. Panel (**b**) is an adaptive directional filtering operator.

**Figure 11 sensors-23-06315-f011:**
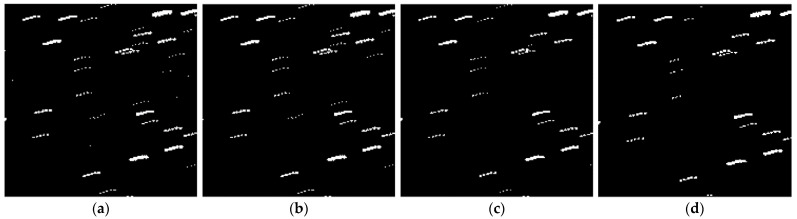
Enlarged local results of different filtering methods. Panel (**a**) represents the image before filter. Panel (**b**) represents the adaptive filtering result. Panel (**c**) represents the improved median filtering result. Panel (**d**) represents the local threshold filtering result.

**Figure 12 sensors-23-06315-f012:**
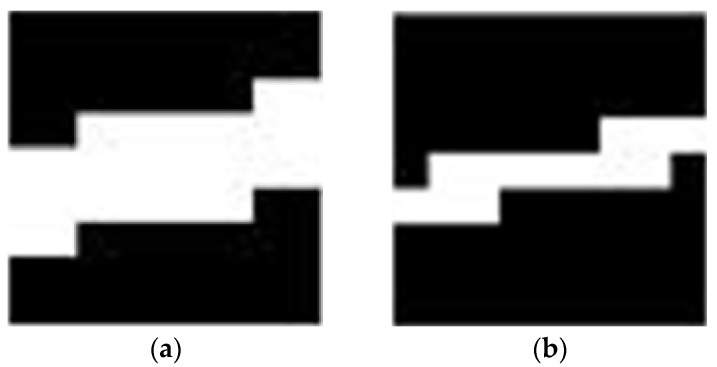
Panel (**a**) is morphological dilation operator. Panel (**b**) is morphological corrosion operator.

**Figure 13 sensors-23-06315-f013:**
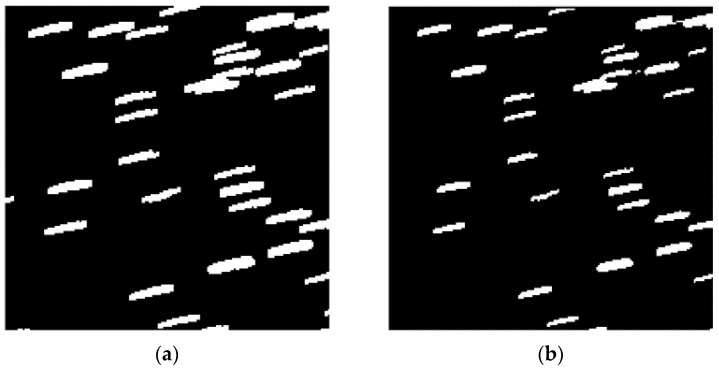
Panel (**a**) shows the local results of morphological expansion. Panel (**b**) shows the local corrosion results after morphological expansion.

**Figure 14 sensors-23-06315-f014:**
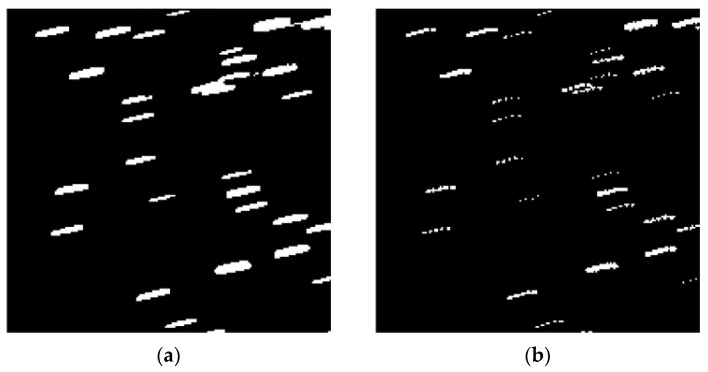
Panel (**a**) shows the local adaptive median filtering graph. Panel (**b**) shows the local final trajectory detection graph.

**Figure 15 sensors-23-06315-f015:**
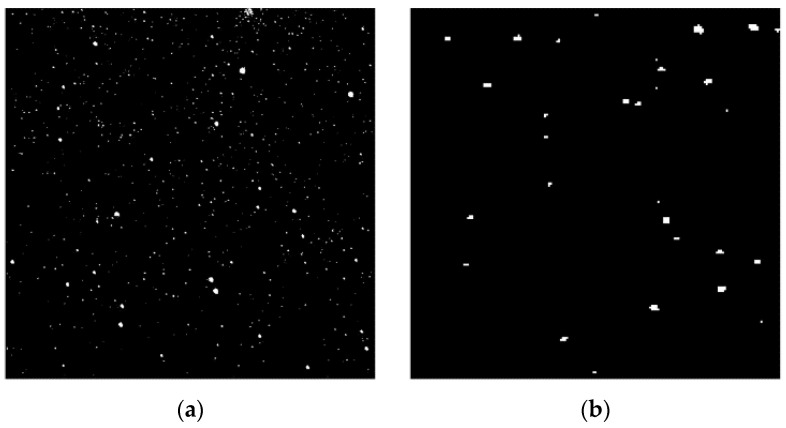
Panel (**a**) shows the fifth frame star positioning image. Panel (**b**) shows a partially enlarged view of panel (**a**).

**Figure 16 sensors-23-06315-f016:**
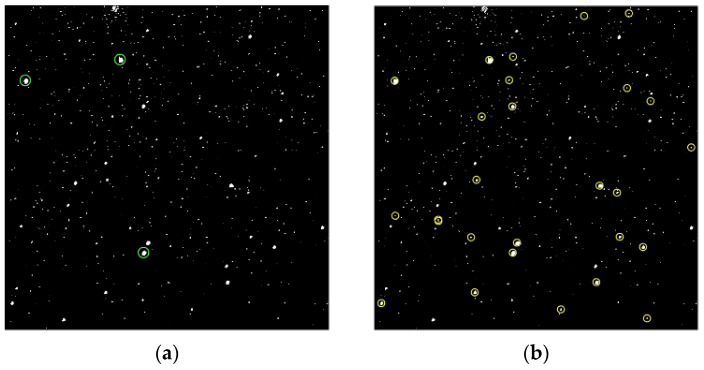
The successfully matched stars are marked in panel (**a**). The successfully identified stars are marked in panel (**b**).

**Figure 17 sensors-23-06315-f017:**
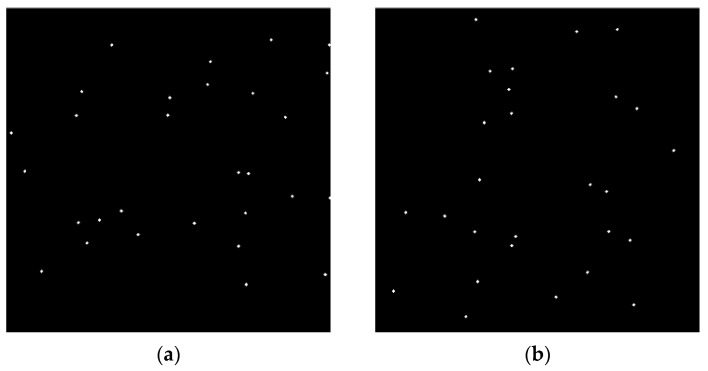
Panel (**a**) represents the ideal imaging of the camera. Panel (**b**) represents the ideal imaging after image rotation correction.

**Figure 18 sensors-23-06315-f018:**
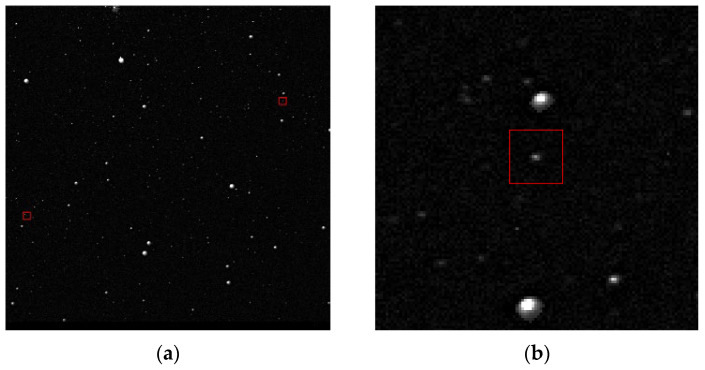
The two weakest stars are marked with red box in panel (**a**). Panel (**b**) is a locally enlarged image of panel (**a**).

**Table 1 sensors-23-06315-t001:** The results of star detection using our method.

SNR	Average Number	Recognition Rate	False-Alarm Rate
1.5	853	97.8%	5.09%
2	655	97.2%	4.66%
3	445	96.5%	3.97%
4	329	94.2%	3.14%

**Table 2 sensors-23-06315-t002:** The results of star detection using MP.

SNR	Average Number	Recognition Rate	False-Alarm Rate
1.5	5010	98.1%	74.3%
2	1476	96.8%	50.1%
3	714	92.6%	44.2%
4	496	88.5%	36.7%

**Table 3 sensors-23-06315-t003:** The results of star detection using IFM.

SNR	Average Number	Recognition Rate	False-Alarm Rate
1.5	2673	97.9%	74.8%
2	854	97.2%	51.4%
3	538	87.8%	42.5%
4	378	37.0%	39.9%

## Data Availability

Not applicable.

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
