# Peer review of "Weak Spatial Target Extraction Based on Small-Field Optical System"

_sensors, 2023, doi:10.3390/s23146315_

Round 1
Reviewer 1 Report
The paper is interesting and meets the journal scope. However, the following points should be addressed before the paper can be accepted for publication.
1. Line 145: Need to add the original image of Figure 1 and need to add annotations to the axes.
2. Line 150: What is the morphological operator and what is its principle for image processing? What is the size of the operator?
3. Line 181: The morphological operator has been improved in the paper, please explain how it has been improved for background correction?
4. Line 193: Need to add annotations to the axes.
5. Line 208: Please explain: Is there no overexposure with an integration time of 1 second, and will the target fly out of the field of view in 1 second? Is it necessary to move the telescope when observing with the telescope?
6. Line 215: How is the SNR defined?
7. Line 230: The author does not seem to understand the relationship between frame rate and exposure time. And this part is not directly related to the final result of image enhancement using 5 frames of images.
8. Line 263: How to calculate the SNR of a single pixel point?
9. Line 266: Please explain the meaning of "full dimensional matched filter" and "two-dimensional matched filter".
10. Line 269: How are the optimal, summation, and maximum projections projected, and what are their principles?
11. Line 269: Is there a connection between "mean projection" and the previous three methods?
12. Line 282: Check English.
13. Line 304: Explain why the maximum projection is more convenient to release the projection.
14. Line 318: Need to add an introduction to the principle of adaptive threshold method.
15. Line 420: "......the value at 2/3 420 positions......", is there a basis for this?
16. Line 443: Equations (14) and (15) need to be presented in detail.
17. Line 454: Check the parameters of the telescope for accuracy.
18. Line 492: Why there is only one star in Figure 18(b).
19. Line 500: Are the stars in the other frames also positioned using equations (14) and (15)?
20. Line 20: Explain the meaning of "average number".
Major revision
Reviewer 2 Report
The authors demonstrate a series of algorithms to detect small objects and their trajectories on stellar images.
The algorithms are clearly described and illustrated. The presented examples are convincing.
What missing is the mathematical analysis of optimality of discussed filtering algorithms. In view of practical orientation of the paper this is, however, not that important.
I guess, the paper can be accepted in the present form.
I'm not qualified to evaluate the quality of English Language. However, I don't feel discomfort reading the paper. I find it written quite well.
Author Response
Thanks for your valuable suggestion. We will consider your proposal in the future research.
Wish you a happy life!
Reviewer 3 Report
The paper is well prepared and organised. The structure is good.
I would only suggest you used past simple tense in the Conclusions as they usually summarize what had been done. Hence, it might sound better if you say "... we proposed"; "this paper proposed", ... and so on.
You also have more than one conclusion in the conclusion section, why not call it CONCLUSIONS !
Also do not use contraction form in formal language.
Lines 462 and 514 ---> can't. Use cannot instead.
See my comments in the Author's part above.
Author Response
Thanks for your valuable suggestion. We have made corresponding revisions in the paper. All revisions to the manuscript have been highlighted.
Wish you a happy life!
Round 2
Reviewer 1 Report
Accept